# ARPNet: Antidepressant Response Prediction Network for Major Depressive Disorder

**DOI:** 10.3390/genes10110907

**Published:** 2019-11-07

**Authors:** Buru Chang, Yonghwa Choi, Minji Jeon, Junhyun Lee, Kyu-Man Han, Aram Kim, Byung-Joo Ham, Jaewoo Kang

**Affiliations:** 1Department of Computer Science and Engineering, Korea University, Seoul 02841, Korea; buru_chang@korea.ac.kr (B.C.); yonghwachoi@korea.ac.kr (Y.C.); mjjeon@korea.ac.kr (M.J.); ljhyun33@korea.ac.kr (J.L.); 2Department of Psychiatry, Korea University Anam Hospital, Korea University College of Medicine, Seoul 02841, Korea; han272@korea.ac.kr; 3Department of Biomedical Sciences, Korea University College of Medicine, Seoul 02841, Korea; chloe_ark@korea.ac.kr; 4Brain Convergence Research Center, Korea University Anam Hospital, Seoul 02841, Korea; 5Interdisciplinary Graduate Program in Bioinformatics, Korea University, Seoul 02841, Korea

**Keywords:** major depressive disorder, antidepressant response prediction, patient representation, neural network

## Abstract

Treating patients with major depressive disorder is challenging because it takes several months for antidepressants prescribed for the patients to take effect. This limitation may result in increased risks and treatment costs. To address this limitation, an accurate antidepressant response prediction model is needed. Recently, several studies have proposed models that extract useful features such as neuroimaging biomarkers and genetic variants from patient data, and use them as predictors for predicting the antidepressant responses of patients. However, it is impossible to utilize all the different types of predictors when making a clinical decision on what drugs to prescribe for a patient. Although a machine learning-based antidepressant response prediction model has been proposed to overcome this problem, the model cannot find the most effective antidepressant for a patient. Based on a neural network, we propose an Antidepressant Response Prediction Network (ARPNet) model capturing high-dimensional patterns from useful features. Based on a literature survey and data-driven feature selection, we extract useful features from patient data, and use the features as predictors. In ARPNet, the patient representation layer captures patient features and the antidepressant prescription representation layer captures antidepressant features. Utilizing the patient and antidepressant prescription representation vectors, ARPNet predicts the degree of antidepressant response. The experimental evaluation results demonstrate that our proposed ARPNet model outperforms machine learning-based models in predicting antidepressant response. Moreover, we demonstrate the applicability of ARPNet in downstream applications in use case scenarios.

## 1. Introduction

Treatment of major depressive disorder (MDD), which is one of the most common mental illnesses, is challenging. The main reason is that it takes a long time (8–12 months) for the effects of an antidepressant to be seen in a patient. It also takes several months for psychiatrists to find that some drugs are ineffective. As a result, only 11–30% of patients reach clinical remission from initial treatment [1,2,3,4]. This limitation of MDD treatment may result in increased risks and treatment costs. To address this limitation, an accurate prediction model that can predict antidepressant responses is needed.

Recently, several studies that extract useful predictors for predicting the antidepressant responses from patient data such as genetic and brain imaging information [5,6] have been proposed. For example, neuroimaging biomarkers [7], genetic variants [8,9,10,11,12], and DNA methylation [13] have been proven to be effective in improving prediction performance. However, it is impossible for even an experienced psychiatrist to utilize all predictors to make a clinical decision on what drugs to prescribe for a patient.

To overcome this limitation, a machine learning-based antidepressant response prediction model [14] has been proposed. Machine-learning based approach can be used to detect high-dimensional patterns of all potential predictors extracted from various patient data. The model of Chekroud et al. [14] extracts useful predictors from the data of the multicenter clinical trial on MDD (STAR*D) [15,16,17,18] and predicts the antidepressant “citalopram” response in MDD patients. Using the Elastic network model [19], the model of Chekroud et al. [14] first selects useful predictors such as sociodemographic features and depression severity checklists from patient data for predicting antidepressant responses. The model then predicts whether a patient will reach clinical remission based on the 25 selected predictors.

However, the above antidepressant response prediction model has some limitations. The prediction model is designed for predicting the response of only one antidepressant, “citalopram” (Limitation 1). In addition, the model only predicts whether the patient is responding to the antidepressant. The model cannot measure the degree of antidepressant response (Limitation 2). This model is limited as it cannot be used for finding the most effective drugs for MDD patients among many antidepressant candidates (Limitation 3).

To address these limitations, we propose the Antidepressant Response Prediction Network (ARPNet) model based on a neural network architecture. Our proposed ARPNet model is designed for real-world antidepressant prescription scenarios where the prescription consists of a combination of one or more antidepressants (related to Limitation 1). When an antidepressant combination and patient data including the demographic, genetic, and MRI information of a patient are given, ARPNet can predict the patient’s Hamilton Depression Rating Scale (HAM-D) score at the next visit (related to Limitation 2). Our ARPNet model is capable of prescribing the most effective antidepressant combination by predicting not only whether the patient will respond to the prescribed antidepressant combination but also the degree of response (HAM-D score) (related to Limitation 3). Moreover, ARPNet trains patient representation vectors that capture patient characteristics, and antidepressant representation vectors that capture the features of antidepressants. The trained patient and antidepressant representation vectors could be used in downstream applications such as drug discovery for new antidepressant development and patient clustering in clinical decision support systems. The vectors can also be used to find similar patients who are likely to respond to the same drugs.

The contributions of our study are summarized as follows.
We propose an antidepressant response prediction network model called ARPNet based on a neural network architecture. Unlike the previous models, our proposed model predicts whether the patient will reach clinical remission from depression, and the degree of antidepressant response. By predicting the degree of antidepressant response, our proposed model can prescribe the most effective antidepressants.ARPNet can be used in a real-world antidepressant prescription scenario where psychiatrists prescribe one or more antidepressants to patients with MDD. ARPNet predicts the antidepressant response to a given combination of antidepressants, which could not be done by previously proposed models.Through the literature-based and data-driven feature selection process, ARPNet extracts useful features from patient data including the demographic, genetic, and MRI information of patients.The trained patient and antidepressant representation vectors of ARPNet could be utilized in various downstream applications such as drug discovery and clinical decision making.

The remainder of this paper is organized as follows. In Section 2, we introduce the data, including the prescription records of patients with MDD, which is used for antidepressant response prediction. We describe the feature selection process which involves extracting useful predictors from patient data. We then describe our proposed ARPNet model which uses selected predictors and the prescription records of patients. In Section 3, we conduct experiments to assess ARPNet, and analyze the experimental results. In Section 4, we conclude the paper.

## 2. Materials and Method

In this section, we first describe the MDD patient data used in our study. Next, we introduce our feature selection process, which involves selecting useful features as predictors for improving performance in antidepressant response prediction. Then, we state the antidepressant response prediction task. Last, we introduce our proposed model ARPNet.

### 2.1. Data Description

In this study, we used the data of 121 patients with MDD collected from Korea University Anam Hospital, Seoul, Korea. All patients who are ethnically Korean were examined by trained psychiatrists using a structured clinical questionnaire. The depression severity of the patients was measured using the 17-item Hamilton Depression Rating (HAM-D 17) scale and 21-item Hamilton Depression Rating (HAM-D 21) scale at every visit. In this study, we employed the HAM-D 17 scale for the predictions on the degree of the antidepressant response. The collected MDD patient data include the demographic, genetic, and MRI information of only patients who consented to the use of their data for this study. Some patients did not consent to the use of certain data, thus the data available for each patient may vary. If one patient’s *i*th feature is missing, we randomly chose one of the *i*th features of all the remaining patients, and used it as the *i*th feature of the patient. Specifically, 67, 71, 96, and 91 patients consented to the use of their demographic, MRI, and genetic information, respectively.

We confirm that the Institutional Review Board of Korea University Anam Hospital approved this study (ED16162). The statistics are summarized in Table 1.

### 2.2. Feature Selection

We used a two-step feature selection process to extract the most useful features from patient data. The first step involved selecting features identified by a literature survey. The second step involved further refining the features identified in the first step using data-driven methods. We did not perform feature selection on demographic information because it already contains features considered useful by psychiatric experts.

#### 2.2.1. Literature-Based Feature Selection

From the MRI and genetic information, we selected the following three literature-based features.

**Neuroimaging biomakers.** Using MRI information, we conducted a literature survey to identify neuroimaging biomarkers of treatment response in patients with MDD [7]. Among the reported biomarkers, we found 94 features in our MRI information. We selected 62 MRI features excluding 32 features which are not found in more than 30% of patients.

**Genetic variants.** Several studies [8,9,10,11,12] have been conducted to identify the effects of genetic variants on antidepressant response. The studies report that variation of 27 significant genes such as 5-HTTLPR, BDNF, and SLC6A4 is associated with antidepressant response.

We found additional genes using biomedical literature-based tools such as BEST [20] and VarDrugPub [21]. BEST, which is a biomedical entity search tool, returns biomedical entities frequently mentioned with query terms in abstracts of published articles. We searched for all the antidepressant prescriptions on BEST, and obtained the Top 23 ranked genes. We also found 12 genes related to antidepressants using VarDrugPub, which is a database containing relationships among genomic variants, drugs, and diseases extracted from the literature.

**DNA methylation.** Genetic variants and epigenetic modification known as DNA methylation [13] are known to affect the treatment for depression. The authors reported that SLC6A4, BDNF, IL11, and MAOA genes and their CpG sites are associated with various antidepressants. We selected 136 CpG sites as DNA methylation features from the four genes.

#### 2.2.2. Data-Driven Feature Selection

We identified and extracted useful features from the literature and filtered unnecessary features using the available APIs. Using Elastic net [19], which is one of the popular regression analysis methods, we selected the more informative features among the extracted features for antidepressant response prediction. The estimates from the elastic net method are defined as follows.
(1)$$\hat{\beta }\equiv \underset{\beta }{\mathrm{argmin}}{(||Y-X\beta ||}^{2}+\lambda_{2}{||\beta ||}^{2}+\lambda_{1}{||\beta ||}_{1}),$$
where *X* is an input vector, *Y* is a true output value and β is the weight of the input vector. We used Elastic net to select the more informative features for treatment of patients with MDD from each of the three feature groups. We concatenated each feature with a one-hot encoded 14-dimensional antidepressant vector as input *X* where the number of antidepressants in our patient data is 14. The difference between HAM-D scores at initial and last visits of patients were used as output *Y*.

For each feature group, we split the data into five subsets and used one subset as the test set and the remaining subsets as the training set to reduce the risk of overfitting. Then, we conducted 10-fold cross validation on the training set to find the hyper parameters of each Elastic net-based feature selection model. After selecting hyper parameters for each feature selection model, we validated the models on the test set and coefficients of the feature. By averaging the coefficients of the features from the five subsets, we selected 20 features with the 10 highest coefficients and the 10 lowest coefficients for each of the feature groups. The coefficients of the selected features are summarized in Table 2.

### 2.3. Problem Statement

When a patient sees a psychiatrist, the psychiatrist may prescribe an antidepressant to the patient based on the data of the patient. To prescribe the most effective antidepressant, the psychiatrist predicts the antidepressant response of the patient. Antidepressant response prediction is defined with this real-world antidepressant prescription scenario in mind.

Based on data *P* including the demographic, MRI, and genetic information of a patient, the current HAM-D score *H* of the patient, and the time interval ΔT until the next visit, antidepressant response prediction models predict the HAM-D score of the patient at the next visit for the given antidepressants *A*. The demographic information includes height, weight, and sex information of patients. Real-valued information such as height and weight is represented with log-scale. Discriminated information such as sex and occupation is encoded with a one-hot vector. The predicted HAM-D score $\hat{Y}$ is computed as follows:
(2)$$\hat{Y}=f\left(P,H,\Delta T,A|\theta \right),$$
where θ is a set of parameters of the prediction model. The parameters were trained in the training phase to minimize the loss that is defined in the next subsection.

### 2.4. ARPNet

Figure 1 shows the architecture of ARPNet. Our source code that implements ARPNet is available at http://github.com/dmis-lab/arpnet. Our ARPNet model consists of the following three components: a patient representation layer, an antidepressant prescription representation layer, and a prediction layer. The notations used in this paper are summarized in Table 3.

#### 2.4.1. Patient Representation Layer

The patient representation layer is designed to capture an abstractive patient feature from patient data. The input $X_{P}\in R^{127+20+20+20+1+1=189}$ of this layer is constructed as the concatenation of the current HAM-D score H∈R and time interval ΔT∈R, which are the useful features selected in Section 2.2, as follows.
(3)$$X_{P}=\left[V_{demo};V_{bio};V_{gene};V_{methyl};H;\Delta T\right],$$
where $V_{demo}\in R^{127}$, $V_{bio}\in R^{20}$, $V_{gene}\in R^{20}$, and $V_{methyl}\in R^{20}$ are the representation of demographic information, neuroimaging biomarkers, genetic variants, and DNA methylation, respectively.

To extract the abstractive patient feature from interactions between selected features, the concatenated input $X_{p}$ is forwarded to a non-linear layer with a non-linear function such as ReLU, tanh, or sigmoid. In this paper, we use ReLU as the non-linear function.
(4)$$V_{P}=\sigma \left(X_{P}\cdot W_{P}+b_{P}\right),$$
where $V_{P}\in R^{d}$ is the *d*-dimensional patient representation vector, and $W_{P}\in R^{189\times d}$ and $b_{P}\in R^{d}$ are the weight matrix and bias term of the patient representation layer, respectively. σ(·) is the ReLU function.

The two major contributions of the computed patient representation vector are summarized as follows. The patient feature representation is automatically trained from patient data to improve the performance in antidepressant response prediction. Moreover, the similarities between patients could be computed by representing patients as real-valued vectors. The similarities between patients are useful in downstream applications such as a clinical decision support system. For example, physicians can find patients who are similar to a given patient using patient representation vectors. The patient representation vectors are trained for antidepressant response prediction. Therefore, patients who have similar vector representations are predicted to respond to the same drugs.

#### 2.4.2. Antidepressant Prescription Representation Layer

A prescription for a patient consists of one or more antidepressants. The prescription representation layer is designed to capture antidepressant features from one antidepressant or the combination of antidepressants. We first associate each antidepressant A∈A with a $d^{\prime }$-dimensional real-valued vector. A is the set of antidepressants in our patient dataset. We denote an antidepressant representation matrix that consists of the $d^{\prime }$-dimensional antidepressant vectors as $E_{A}\in R^{\left|A\right|\times d^{\prime }}$. The prescription is represented as a one-hot vector as follows.
(5)$$X_{A}=\left[x_{A_{1}},x_{A_{2}},\ldots ,x_{A_{|}A|}\right],$$
(6)$$x_{A_{i}}=\left\{\begin{array}{cc} 1, & \mathrm{if}\;\mathrm{the}\;i\mathrm{th}\;\mathrm{antidepressant}\;\mathrm{is}\;\mathrm{in}\;\mathrm{the}\;\mathrm{prescription},\;\\ 0, & \mathrm{otherwise}. \end{array}\right.$$

Finally, the prescription representation is computed as follows.
(7)$${\hat{X}}_{A}=E_{A}\left(X_{A}\right),$$
(8)$$V_{A}=Sum\left({\hat{X}}_{A}\right),$$
where $E_{A}$ is the antidepressant representation look-up function, and ${\hat{X}}_{A}$ denotes the representations of the drugs in the prescription $X_{A}$. $V_{A}\in R^{d^{\prime }}$ is the sum of the antidepressant representations in ${\hat{X}}_{A}$ and is the final antidepressant prescription representation that is utilized in the prediction layer described in the next subsection.

#### 2.4.3. Prediction Layer

The prediction layer is a linear regression approach based on the patient representation and the prescription representation. An input of this layer is constructed by concatenating the patient representation and the prescription representation as follows.
(9)$$X=[V_{P};V_{A}],$$
where $X\in R^{d+d^{\prime }}$ is the input of the prediction layer. Then, the input *X* is forwarded to the regression layer as follows.
(10)$$\hat{Y}=X\cdot W+b,$$
where $W\in R^{(d+d^{\prime })\times 1}$ and $b\in R^{1}$ are the weight matrix and the bias term of the regression layer, respectively.

To optimize the parameters θ of our proposed model, we define the loss function of our model using Mean Square Error (MSE) as follows.
(11)$$MSE=\frac{1}{\left|D\right|}\sum_{i=1}^{\left|D\right|}{\left(Y_{i}-{\hat{Y}}_{i}\right)}^{2},$$
where $Y_{i}$ is a true HAM-D score of the *i*th prescription record data. The model parameters are randomly initialized with a Gaussian distribution and trained by minimizing the MSE using the Adam optimizer [22] with a learning rate 0.001 and a batch size 16. To address the overfitting problem, we apply dropout [23] and L2 regularization. Based on cross-validation, which is described in the next section, we find the optimal hyperparameter setting of our proposed ARPNet with grid search.

We implemented our proposed model using TensorFlow v1.6.0 (https://www.tensorflow.org) and Python v2.7.12 (https://www.python.org). Our experiments were conducted on Ubuntu 16.04.5 with two TITAN X (Pascal) GPUs.

## 3. Experiments and Results

In this section, we evaluate our proposed ARPNet model by comparing it with several machine learning-based baseline models. We conducted experiments to answer the following research questions:

**RQ1**. Does ARPNet outperform the baseline models in predicting the degree of antidepressant response?

**RQ2**. Can ARPNet predict whether a patient reaches clinical remission, which is a task proposed in previous work?

**RQ3**. Is ARPNet useful for prescribing the most effective antidepressant combination to patients?

**RQ4**. How can we use the trained patient and antidepressant representation vectors in downstream applications?

We describe the experimental setting and report the evaluation results on the prediction of the degree of antidepressant response (Section 3.1) and the prediction of whether a patient reaches clinical remission with the given antidepressant (Section 3.2). We introduce the use case scenarios of our ARPNet model. (Section 3.3 and Section 3.4).

### 3.1. Task 1: Prediction of the Degree of Antidepressant Response

#### 3.1.1. Dataset Preparation

The prescription records of patients contain the HAM-D scores of the patients measured at the initial visit and one, four and eight weeks (or six months) after the initial visit. Some prescription records are missing the prescription record for the week in which patients did not visit. The data d=(P,A,H,ΔT) used in this experiment were generated from all pairs of two consecutive visit records ($r_{i}$ and $r_{i+1}$) of a patient when the prescription records $R_{P}=\left\{r_{1},r_{2},\ldots ,r_{|R_{P}|}\right\}$ of the patient are given. ΔT is the interval between the *i*th and i+1th visits. Our dataset D consists of 273 data generated from 395 records of 121 patients. The randomly sampled 10% of data were used as the test set, and the remaining data were used as the training set. The statistics of the constructed dataset *D* are summarized in Table 4.

#### 3.1.2. Baselines

For the qualitative evaluation, we compares the performance of ARPNet with that of the following six machine learning baseline models: Support Vector Machine Regressor with a linear kernel (Linear SVR), Ridge Regressor, Gradient Boosting, Multi-Layer Perceptron (MLP) Regressor, K-Nearest Neighbors Regressor, and Random Forest Regressor. We concatenates the predictor features selected in Section 2.2, one-hot encoded prescribed antidepressants, current HAM-D score of a patient, and the interval between visits as an input of the machine learning baseline models.

To find the optimal hyper-parameters for our proposed ARPNet model and the baseline models, we performed five-fold cross validation on the training set.

#### 3.1.3. Metric

Antidepressant response prediction is a regression task. To assess the performance of the models, we employed Root Mean Square Error (RMSE) and R-squared, which are the most common evaluation metrics used for regression models. RMSE is computed as follows.
(12)$$RMSE=\sqrt{\frac{1}{|D_{test}|}\sum_{i=1}^{|D_{test}|}{\left(y_{i}-{\hat{y}}_{i}\right)}^{2}}.$$

R-squared is computed as follows.
(13)$$R-squared=1-\frac{\sum_{i=1}^{|D_{test}|}{\left(y_{i}-{\hat{y}}_{i}\right)}^{2}}{\sum_{i=1}^{|D_{test}|}{\left(y_{i}-\bar{y}\right)}^{2}},$$
where $\bar{y}$ denotes the average value of actual HAM-D scores in our dataset $D_{test}$. A lower RMSE indicates that the model achieves better prediction performance. A higher R-squared also indicates better performance.

#### 3.1.4. Results

To reduce the randomness of the machine learning-based models that randomly initialize their parameters, we performed the training and evaluation five times using the optimal hyper-parameters obtained from cross-validation. We then report the mean values of the performance scores.

The left side of Table 5 shows the evaluation results on the prediction of the degree of antidepressant response (Task 1). Our model ARPNet outperforms all the baseline models in terms of RMSE and R-squared. Specifically, ARPNet obtains 27.9%, 12.2%, 25.2%, 14.9%, and 12.8% performance improvements in terms of RMSE over LinearSVR, Ridge, Gradient Boosting, K-Nearst Neighbors, and Random Forest, respectively. ARPNet also obtains 372.9%, 31.5%, 174.0%, 44.4%, and 34.1% performance improvements in terms of R-squared over Support Vector Machine Regressor with a linear kernel (Linear SVR), Ridge Regressor, Gradient Boosting, Multi-Layer Perceptron (MLP) Regressor, K-Nearest Neighbors Regressor, and Random Forest Regressor, respectively.

These observations demonstrate that ARPNet is more accurate than the baseline models in predicting the degree of antidepressant response (Answer to RQ1).

### 3.2. Task 2: Prediction of the Clinical Remission of Patients

Prediction of the clinical remission of patients is a binary classification task. We assumed that a patient responded to the prescribed antidepressant if the HAM-D score of the patient measured at their last visit is less than half the HAM-D score measured at the initial visit. A patient’s response to an antidepressant is calculated as follows.
(14)$$Y_{P}^{\prime }=\left\{\begin{array}{cc} 1, & \mathrm{if}\;\frac{H_{|R_{P}|}}{H_{1}}\leq 0.5\;\\ 0, & \mathrm{otherwise}, \end{array}\right.$$
where $Y_{P}^{\prime }$ is the label of the patient *P* and $H_{i}$ is the HAM-D score of the patient at the *i*th visit. Prediction models predict the clinical remission of patients using Algorithm 1.

**Algorithm 1** Prediction of the clinical remissions of patients.
**Input:***P*: Patient, $R_{P}$: Prescription records {$r_{1}$, $r_{2}$, ⋯, $r_{|R_{P}|}$} of *P*
**Output:**${\hat{Y}}_{P}^{\prime }$: Predicted clinical remission of patient *P*.
1:
**procedure**
2:    Initialize current HAM-D score $H\leftarrow H_{1}\in r_{1}$3:    **for** i **in** [1, |$R_{P}$|) **do**4:        *i*th record’s HAM-D score $H_{i}\leftarrow H$5:        Predict (*i*+1)-th record’s HAM-D score ${\hat{H}}_{i+1}$ from $r_{i}$6:        Current HAM-D score $H\leftarrow {\hat{H}}_{i+1}$7:    **end for** 8:    |$R_{P}$|-th record’s HAM-D score $H_{|R_{P}|}\leftarrow H$  9:    **if**
$\frac{H_{|R_{P}|}}{H_{1}}\leq 0.5$
**then** 10:        **Return**
${\hat{Y}}_{P}^{\prime }$ = 111:    **else**12:        **Return**
${\hat{Y}}_{P}^{\prime }$ = 013:    **end if**14:
**end procedure**



#### 3.2.1. Dataset Preparation

Using Equation (Equation 14), we constructed the dataset $D^{\prime }$ which contains 121 data from the prescription records of 121 patients. As in Task 1, 10% of the data were randomly sampled and used as the test set, and the remaining data were used as the training set. To find the optimal hyper-parameters for the prediction models, we performed five-fold cross validation on the training set. The statistics of the constructed dataset $D^{\prime }$ are summarized in Table 4.

#### 3.2.2. Metric

To evaluate performance on the prediction of clinical remission, we employed the following five evaluation metrics that are widely used in binary classification tasks: Sensitivity, Specificity, Precision, F1-score, and Accuracy. Sensitivity and Specificity are computed as follows,
(15)$$Sensitivity=\frac{TP}{TP+FN},\;Specificity=\frac{TN}{TN+FP},$$
where TP, TN, FP, and FN refer to true positive, true negative, false positive, and false negative, respectively. Precision and F1-score are computed as follows,
(16)$$Precision=\frac{TP}{TP+FP},\;F1-score=\frac{2TP}{2TP+FN+FP}.$$

Accuracy is computed as follows,
(17)$$Accuracy=\frac{TP+TN}{TP+TN+FP+FN}.$$

For all the evaluation metrics, the higher is the score, the better is the performance of a model on prediction.

#### 3.2.3. Results

We performed the training and evaluation five times using the optimal hyper- parameters to reduce the randomness of the machine learning-based prediction models. We report the average of the five evaluation results for each model. The right side of Table 5 shows the evaluation results on the prediction of the clinical remission of patients (Task 2). Our ARPNet model outperforms all the baseline models on all the evaluation metrics (Answer to RQ2). ARPNet achieves performance improvements of at least 40.0%, 74.4%, 86.0%, and 57.1% in Specificity, Precision, F1-score, and Accuracy, respectively, over the best baseline model. In terms of Sensitivity, ARPNet achieves performance improvements of 100% over all the baselines except for K-Nearst Neighbors Regressor. As in the evaluation result of Ridge Regressor, sensitivity generally shows a negative correlation between specificity and precision. Nevertheless, ARPNet obtains performance improvements in all the evaluation metrics.

We observe that ARPNet achieves a greater performance improvement in Task 2 than in Task 1. Figure 2 shows two representative cases: (1) the patient improves over time; and (2) the condition of the patient worsens after four weeks of treatment. In both cases, ARPNet accurately predicts the outcome of the patient as the predicted HAM-D score of the patient is almost the same as their actual HAM-D score. On the other hand, K-Nearest Neighbors and Gradient Boosting were unable to accurately predict the HAM-D scores of patients. Their prediction performance decreased as the interval between visits increased. As it takes a long time (more than eight weeks) to treat MDD, inaccurate antidepressant response predictions can result in ineffective prescriptions and increased risks.

### 3.3. Use Case Scenario: Prescribing the Most Effective Antidepressant

As ARPNet can predict the degree of the antidepressant response, ARPNet can be used to prescribe the most effective antidepressants. We conducted an additional qualitative evaluation to show the applicability of ARPNet. We computed the predicted HAM-D scores of all the possible antidepressant candidates. The possible antidepressant candidates include 14 single antidepressants and 91 combinations of any two antidepressants. Figure 3 shows the result of a sample patient as an example. In the figure, the tables list the predicted HAM-D score of each antidepressant candidate for Patient A at their visit. The green line indicates the change in the actual HAM-D score of Patient A. The red line indicates the change in the HAM-D score predicted by ARPNet for the prescribed antidepressant MIRTAZAPINE. The blue line denotes the change in the HAM-D score when Patient A at the initial visit is prescribed the antidepressant combination of MIRTAZAPINE and MILNACIPRAN which is predicted to be the most effective. In this phase, it is predicted that MIRTAZAPINE with MILNACIPRAN (Rank 1) is more effective than MIRTAZAPINE alone (Rank 3). At the visit in Week 8, the antidepressant combination of ESCTALOPRAM and AMITRIPTYLINE, which is predicted to be the most effective antidepressant, is prescribed. The cyan line denotes the change in the HAM-D score when the psychiatrist changes the prescription to the most effective antidepressants predicted at the visit in Week 8. As shown in this example, the HAM-D score of the patient decreases faster than the actual HAM-D score by prescribing the most effective antidepressants at every visit.

These observations demonstrate that ARPNet can compute the predicted HAM-D scores of all the possible antidepressant candidates to prescribe the most effective antidepressant, which can lead to more effective treatment and reduced costs and risks (Answer to RQ3).

### 3.4. Use Case Scenario: Using Pre-Trained Representation Vectors

Through the patient representation layer, ARPNet trains the patient representation vectors to capture high-dimensional patterns for improving performance in antidepressant response prediction. By representing the patients as real-valued vectors, we can compute the antidepressant response-aware similarity between patients. This antidepressant response-aware similarity can be used for precision prescription in evidence-based medicine [24,25]. Figure 4 shows an example of using patient representation vectors in clinical decision support systems. Patients in the prescription record database are represented as patient representation vectors, using patient data and prescription records. Based on their representation vectors, the patients are clustered using a clustering method that computes the antidepressant response-aware similarity between the patients. When a new patient’s data are given, the patient’s representation vector is computed using the patient representation layer of ARPNet. Clinical decision support systems compute the similarities between the patient and patients in the cluster in which the patient is likely to be included, and provide psychiatrists with the prescription records of patients very similar to those of the patient for evidence-based medicine. Psychiatrists prescribe the most effective antidepressant to their patient by referring to the antidepressant response records of similar patients.

This use case demonstrates that pretrained patient representation vectors can be utilized as clinical evidence in downstream applications for precision medicine (Answer to RQ4).

## 4. Conclusions

In this paper, we propose ARPNet, which is a new antidepressant response prediction network model for major depressive disorder. ARPNet utilizes several useful features from patient data including the demographic, MRI, and genetic information of patients. Based on a literature survey, features such as neuroimaging biomarkers were extracted from MRI information. In addition, genetic variants and DNA methylation were extracted from genetic information using biomedical literature-based tools. To make the extracted features more predictive, we filtered them by a data-driven feature selection process. Based on a neural network architecture, ARPNet predicts not only whether patients will respond to an antidepressant, but also the degree of the antidepressant response. ARPNet considers a real-world antidepressant prescription scenario where psychiatrists prescribe one or more antidepressants to patients with MDD. In the experiments, ARPNet outperformed the machine learning-based baseline models in both the prediction of the degree of antidepressant response and the prediction of clinical remission of patients. Finally, we used real-world use cases to show the applicability of ARPNet in downstream applications. In future work, we plan to further evaluate ARPNet on real patients.

## Figures and Tables

**Figure 1 genes-10-00907-f001:**
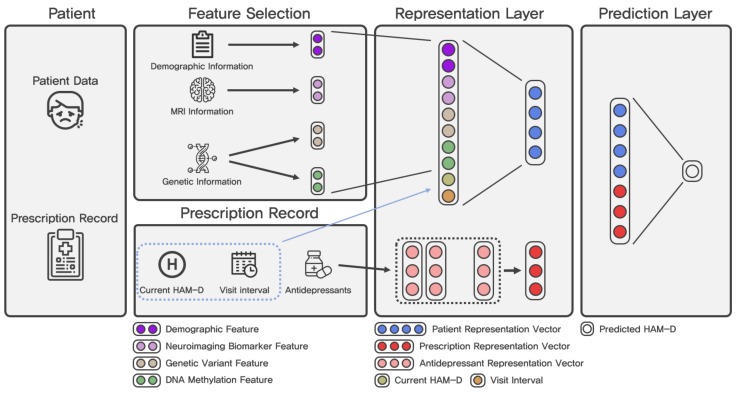
Architecture of our proposed ARPNet model.

**Figure 2 genes-10-00907-f002:**
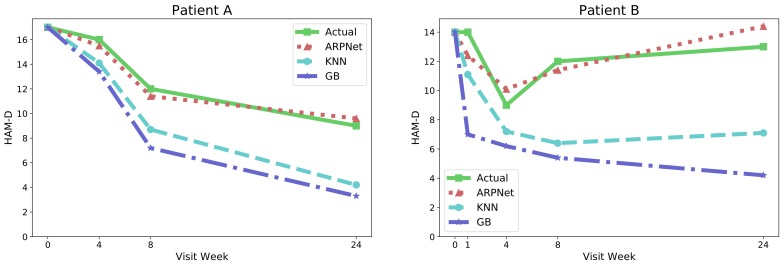
Examples of predicting the clinical remission of patients. KNN and GB denote the K-Nearst Neighbors and Gradient Boosting Regressors, respectively.

**Figure 3 genes-10-00907-f003:**
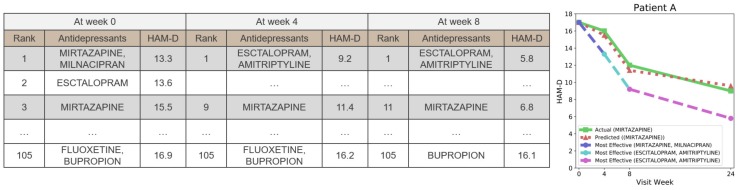
The use case where the most effective antidepressant is prescribed. The first tables show the ranked predicted HAM-D scores of Patient A at their first visit. A green line indicates an actual change in the HAM-D score of Patient A. A red line indicates a change in the HAM-D score predicted by ARPNet for the prescribed antidepressant MIRTAZAPINE. The blue line indicates the changes in the predicted HAM-D score when Patient A is prescribed MIRTAZAPINE and MILNACIPRAN, which are predicted to be the most effective at first visit. The cyan and pink lines indicate the predicted HAM-D score changes of Patient A when the antidepressants are changed to ESCITALOPRAM and AMITRIPTYLINE. ESCITALOPRAM and AMITRIPTYLINE are the most effective antidepressants predicted by ARPNet in Weeks 4 and 8.

**Figure 4 genes-10-00907-f004:**
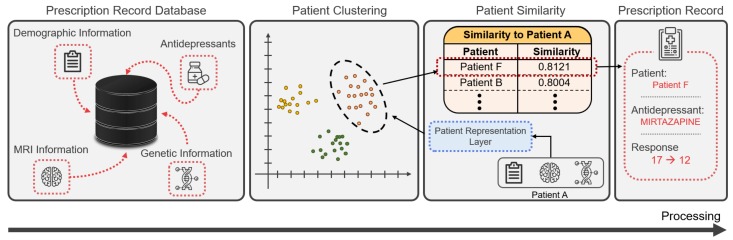
A use case of clinical decision support systems using the similarity between patients.

**Table 1 genes-10-00907-t001:** Selected features.

Neuroimaging Biomarkers		Genetic Variants		DNA Methylation		
Feature Name	Coefficient	Feature Name	Coefficient	Feature Name	Gene Name	Coefficient
lh_G_front_inf_Orbital_thickness	−5.0191	HTR1A:p.Ala155Gly	−7.2980	cg03829016	SLC6A4	−89.1863
lh_G_cingul_Post_dorsal_thickness	−3.9096	PAPLN:p.Gly838Glu	−7.0918	cg17075252	BDNF	−64.8215
lh_G_front_middle_thickness	−3.2466	TPH2:p.Arg225Gln	−6.3323	cg06373684	SLC6A4	−62.5500
lh_G_cingul_Post_ventral_thickness	−3.1023	ABCB1:p.Ala599Thr	−5.8213	cg26741280	SLC6A4	−58.5697
rh_G_insular_short_thickness	−2.5389	TNFSF14:p.Ala77Val	−4.1921	cg15462887	BDNF	−56.0164
rh_G_and_S_cingul_Mid_Ant_thickness	−2.2651	SCN5A:p.Pro1090Leu	−3.0044	cg18354203	BDNF	−50.8337
lh_G_and_S_cingul_Mid_Post_thickness	−1.8986	OPRM1:p.Gln402His	−2.8859	cg10241426	SLC6A4	−50.3451
lh_G_oc_temp_lat_fusifor_thickness	−1.8417	CYP2D6:p.Gly169Arg	−2.4712	cg06260077	BDNF	−47.2429
rh_G_front_middle_thickness	−1.5880	MTHFR:p.Ile248Val	−2.4050	cg07919246	BDNF	−46.7545
lh_G_insular_short_thickness	−1.5748	BDNF:p.Arg109Gln	−2.4010	cg15014679	BDNF	−38.1958
rh_G_cingul_Post_ventral_thickness	1.2208	BMP7:p.Arg154Gln	3.3401	cg10558494	BDNF	43.3355
rh_G_parietal_sup_thickness	1.4562	FKBP5:p.Val437Phe	3.3898	cg16737991	IL11	45.6990
lh_G_and_S_cingul_Mid_Ant_thickness	1.9680	IL11:p.Arg98Pro	3.4588	cg04672351	BDNF	48.9424
rh_G_and_S_cingul_Mid_Post_thickness	2.1854	OPN1SW:p.Ile302Val	3.5439	cg06961290	SLC6A4	50.3463
lh_G_oc_temp_med_Lingual_thickness	2.7142	TPH2:p.Ser41Tyr	3.7021	cg17882499	BDNF	59.1599
rh_G_front_inf_Opercular_thickness	3.3577	DRD4:p.Ala84Thr	3.7587	cg07238832	BDNF	70.9359
rh_G_cingul_Post_dorsal_thickness	4.3347	CDH17:p.Tyr79Cys	4.4870	cg11241206	BDNF	87.2378
rh_G_front_inf_Triangul_thickness	5.1329	TPH1:p.Arg248*	4.5572	cg07159484	BDNF	96.8888
rh_G_front_inf_Orbital_thickness	6.6912	RORA:p.Pro15Leu	7.2040	cg05016953	SLC6A4	111.0701
lh_G_parietal_sup_thickness	9.0480	PML:p.Arg755His	8.3033	cg01636003	BDNF	181.9174

**Table 2 genes-10-00907-t002:** Statistics of feature selection data.

Feature	# of Features at First-Step	# of Features at Second-Step
Demographic information	127	127
Neuroimaging biomarkers	62	20
Genetic variants	156	20
DNA methylation	136	20

**Table 3 genes-10-00907-t003:** Notations.

Notation	Description
Patient Representation Layer	
*P*, P	Patient and set of patients
$X_{P}$	Input vector of the representation layer
$H_{P}$	Current HAM-D score of the patient
$V_{demo}$	Demographic information of the patient
$V_{bio}$	Neuroimaging biomarkers of the patient
$V_{gene}$	Genetic variants of the patient
$V_{methyl}$	DNA methylation of the patient
$\Delta T_{P}$	Visit interval between two consecutive visits of the patient
$W_{P}$, $b_{P}$	Weight matrix and bias term of the patient representation layer
$V_{P}$	Patient representation vector
Antidepressant Prescription Representation Layer	
*A*, A	Antidepressant and set of antidepressants
$E_{A}$	Antidepressant representation matrix
$X_{A}$	Input of the antidepressant prescription representation layer
${\hat{X}}_{A}$	Looked up antidepressant prescription representation
$V_{A}$	Antidepressant representation vector
Prediction Layer	
*X*	Input vector of the prediction layer
*Y*, $\hat{Y}$	True and predicted HAM-D score
*W*, *b*	Weight matrix and bias term of the prediction layer

**Table 4 genes-10-00907-t004:** Statistics of datasets used in experimental evaluation.

Statistics	Dataset D	Dataset $D^{\prime }$
# of data	273	121
# of training data	243	108
# of test data	30	13

**Table 5 genes-10-00907-t005:** Experimental evaluation results. The performance improvement of ARPNet over the baseline with the best performance is reported in the *Improv.* row.

Model	Task 1		Task 2				
	RMSE	R-Squared	Sensitivity	Specificity	Precision	F1-Score	Accuracy
Linear SVR	4.5774	0.1168	0.3200	0.4250	0.1662	0.2167	0.3846
Ridge Regressor	3.7609	0.4199	0.7417	0.2833	0.4300	0.4509	0.4400
Gradient Boosting Regressor	4.4122	0.2016	0.4000	0.6250	0.4000	0.4000	0.5385
K-Nearst Neighbors Regressor	3.8805	0.3824	0.2000	0.5000	0.2000	0.2000	0.3846
Random Forest Regressor	3.7867	0.4118	0.2000	0.4750	0.1717	0.1835	0.3692
ARPNet	**3.3022**	**0.5523**	**0.8000**	**0.8750**	**0.8000**	**0.8000**	**0.8462**
Improv.	12.2%	31.5%	7.9%	40.0%	74.4%	86.0%	57.1%

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
