# Peer review of "ARPNet: Antidepressant Response Prediction Network for Major Depressive Disorder"

_genes, 2019, doi:10.3390/genes10110907_

Round 1

Reviewer 1 Report

The authors developed a deep learning model, called ARPNet, to predict the response of patients about various antidepressants. Based on the current patients' state and HAM-D score, and visit interval, the model predicts the degree of antidepressant response. 

Manuscript Comments:

- The authors mentioned that if the value of one feature of a patient is missing, it is randomly chosen from values of other patients. This process may have a bad effect by distorting the feature data. When a patient only consents to the use of MRI information and the other information is obtained from the other patients, can the patient representation layer correctly represent the state of the patient? Authors should mention the effect of the random selection process for missing values.

- The program was implemented and benchmarked using only one dataset, which may not be enough for accurate evaluation of the model. Authors should evaluate their program using another dataset.

- Also, the performance should be evaluated by comparing with different antidepressant response predicting programs. For example, authors mentioned a model that can predict response about citalopram. Why don’t authors compare the performances of APRNet and the model about the response of citalopram?

- The authors should mention the detail of model structures and training, such as the count of nodes in each layer, how authors decide to when to stop model training. 

Minor Comments:

- Line 228, Authors should fix “RMSE” to “R-squared”

Program Comments:

- In the github page, authors should refer to required Python libraries for ARPNet.

- The instruction about how to run the program using new dataset is missing in the documentation page.

Author Response

We appreciate your comments.

Q1.

>> Good point. We agree with your comment. We agree with your comment. For missing features, we randomly sampled features of other patients. We have tried to sample features based on various data imputation methods. Based on auto-encoder [Rubinsteyn et al., 2016], we imputed the missing data, and then trained prediction models. However, this method did not improve the evaluation performance. We guess that our dataset is insufficient for training the auto-encoder properly. We also designed the imputation process with Hot deck imputation method [Myers et al., 2011] based on Canonical Correlation Analysis (CCA) and Pearson Correlation (PC). However, this method did not contribute to improving performance. Therefore, we choose the simplest method which is random sampling. We agree this approach is not a best choice, but we hope that our proposed ARPNet contributes to the related research areas.

Q2.

>> We found the other two datasets [Chekroud et al. 2016] related to our study. We have tried to obtain the datasets, but we could not obtain the datasets due to privacy issues. Therefore, we release our source code to be utilized as a benchmark antidepressant response prediction model that performs on their own datasets in future research.

Q3.

>> In our MDD patient dataset, only 20 patients are prescribed with "Citalopram". The previously proposed machine learning-based model [Chekroud et al. 2016] was not trained properly because the number of patients is too small. Therefore, we did not employ the previously proposed machine learning-based model.

Q4.

>> We revised the manuscript to include the process of finding the optimal hyperparameters.

Minor Comments:

>> We revised the manuscript.

Program Comments:

>> We added the required Python libraries. However, we cannot release our new dataset due to the privacy issues.

Reviewer 2 Report

In this work, the authors developed a deep learning model for predicting antidepressant response. They compared the prediction performance of their new method, ARPNet, with those of other conventional methods, and concluded that their deep learning based prediction method performed the best. The manuscript is relatively well written, but there are many places where clearer explanations are needed. In terms of methods and results, there are several issues that must be addressed properly.
- L91: The number of patients is only 121, which seems too small considering that the number of features initially considered in this study is 481 even if they reduced the number features by the feature selection procedure. The over-training must be the primary concern for this study, but it is not clear how they did that. In many cases similar to this study, cross validation procedure is not enough.
- - L99: “If one patient’s i-th feature is missing, we randomly choose one of the i-th features of all the remaining patients, and use it as the i-th feature of the patient. Specifically, 67, 71, 96, and 91 patients consented to the use of their demographic, MRI, and genetic information, respectively.”: Randomly choosing missing data seems to be dangerous. Why not using the averages?
- It is not clear to me how all the features are represented. Some of them are quantitative, and some features such as genomic features are nominal variables. According to Line 155, V_gene = R^20, indicating that gene features are real-valued, but how?
- “The randomly sampled 10% of data is used as the test set” in 3.1.1. Dataset Preparation: How was the test set prepared? Data points from the same patients should not be in both training set and test set, that is, all data from a single patient should be in either training set or test set, not in both sets.
- Table 4: What is D’?
- Eq. (7) &(8) in 2.4.2. Antidepressant Prescription Representation Layer section: V_A= Sum(X^_A)? How is this justified? Is E_A matrix a trainable embedding lookup table?
- Table 5: Task 1 and Task 2 are closely related, but ARPNet performed much better on Task 2 than the other methods (accuracy 0.84 vs. 0.37 by random forest), but not so much on Task 1 (RMSE 3.30 vs. 3.79 by rf). How is that possible?
- Algorithm 1: What is optimized? Is it RMSE or Y^’_p? What is the loss function? Are they same for Task 1 and Task 2?

Author Response

We appreciate your comments.

Q1.

>> Good point. We agree with your comments about the overfitting problem. To alleviate the overfitting, we applied l2-regularization to all the trainable parameters and dropout to all the fully-connected layers. We revised the manuscript to include the detailed process.

Q2.

>> Good point. We agree with your comment. For missing features, we randomly sampled features of other patients. We have tried to sample features based on various data imputation methods. Based on auto-encoder [Rubinsteyn et al., 2016], we imputed the missing data, and then trained prediction models. However, this method did not improve the evaluation performance. We guess that our dataset is insufficient for training the auto-encoder properly. We also designed the imputation process with Hot deck imputation method [Myers et al., 2011] based on Canonical Correlation Analysis (CCA) and Pearson Correlation (PC). However, this method did not contribute to improving performance. Therefore, we choose the simplest method which is random sampling. We agree this approach is not a best choice, but we hope that our proposed ARPNet contributes to the related research areas.

Q3.

>> As shown in Table2, except for demographic features, the remaining features are represented as a real-valued vector. we revised the manuscript to include the detailed description of features used in this paper.

Q4

>> We use the different evaluation protocols for task1 and task2 respectively. Task1 is prediction of the degree of the antidepressant response given the current HAM-D score. In this task, since this prediction is used at every visit of patients, data points could be included in both training and test sets. On the other hand, task2 is the prediction of the clinical remission of patients. This prediction is used at the initial visit of patients. Therefore, in this task, all patients are included in either the training set or test set.

Q5.

>> D' is the dataset used in task 2. It is mentioned at Line 237.

Q6.

>> Yes. As you mentioned in the comment, E_A is an antidepressant embedding matrix. We revised the manuscript to reduce the confusion.

Q7.

>> In task 2, the predicted HAM-D score at time t is used as input of the prediction of HAM-D score at time t+1. Therefore, the bigger difference between true HAM-D score and predicted HAM-D score at time t makes the bigger difference at time t+1. As a result, our proposed model is much better than baselined models on task 2 because our proposed model is better on task 1.

Q8.

>> Algorithm 1 describes the procedure of prediction of the clinical remission of patients (task 2) rather than the description of the optimization procedure of our proposed model.

Reviewer 3 Report

This paper developed a method ARPNET to predict antidepressant response. ARPNET integrated neuroimaging markers, genetic variants, and DNA methylation, which are known to be related to antidepressant response. By representing a patient’s information and antidepressant information in the vector space, ARPNET outperformed other machine learning methods using one-hot presentation. I think that this research is promising for the use of antidepressants and a good example of application of the current machine learning techniques.

Page 6) What is the size of A (the number antidepressants used)? Page 8) In the subsection 3.1.2 Baselines, what are the selected predictor features? Are they the same as Eq. (3)? How did you select and represent features? Make it clear.

Author Response

We appreciate your comments.

Q1.

>> Yes. The size of A is the same as the number for antidepressants in our dataset. We revised the manuscript to make it clear.

Q2.

>> Yes. The selected predictor features are in Eq. (3). The predictor features were selected in Section 2.2 feature selection. To be consistent in using the terms, we revised the manuscript.

Reviewer 4 Report

In this work, authors proposed a novel computational model to predict antidepressant response prediction network for major depressive disorder. The proposed model can also able to predict the degree of antidepressant response and can prescribe the most effective antidepressant. The manuscript is well written and each steps in the method development are clearly explained. I have the following comments

In the "Data Description" section, authors mentioned that if a parameter of a patient A is missing/not able to include, the same parameter from other patients will be used for patient A.  How author things this is meaningful ? How this data will provide better discrimination between patients? More clear explanation should be included.

       2. All the patients included in this study are ethnically Koreans. As the data set is not diverse, will this method applicable for patients from other regions?

3. Authors mentioned that randomly selected 10% of data used as test set. How many times author repeated this? in other words, if author randomly select different 10% of data as test set, will the results are same or different?

4. Most of tables are included before the discussion of relevent data in the text which should be corrected

5. Equations for sensitivity, specificity, accuracy, etc, should be included

Author Response

We appreciate your comments.

Q1.

>> Good point. We agree with your comment. For missing features, we randomly sampled features of other patients. We have tried to sample features based on various data imputation methods. Based on auto-encoder [Rubinsteyn et al., 2016], we imputed the missing data, and then trained prediction models. However, this method did not improve the evaluation performance. We guess that our dataset is insufficient for training the auto-encoder properly. We also designed the imputation process with Hot deck imputation method [Myers et al., 2011] based on Canonical Correlation Analysis (CCA) and Pearson Correlation (PC). However, this method did not contribute to improving performance. Therefore, we choose the simplest method which is random sampling. We agree this approach is not a best choice, but we hope that our proposed ARPNet contributes to the related research areas.

Q2.

>> We agree with your comment. Our patient dataset consists of only ethnically Korean patient’s data collected from Korea University Anam Hospital. We found the other two datasets [Chekroud et al. 2016] related to our study. We have tried to obtain the datasets, but we could not obtain the datasets due to privacy issues. Therefore, we release our source code to be utilized as a benchmark antidepressant response prediction model that performs on their own datasets in future research.

Q3.

>> We sampled 10% of data as test sets for two tasks respectively. If we also sample validation sets separately, the size of the training set is too small to train prediction models. Therefore, we employ five-fold cross-validation to find optimal hyperparameters. And then, we repeated the above process five times to reduce the randomness of the machine learning-based prediction models.

Q4.

>> We adjusted the position of tables in revised manuscripts.

Q5.

>> We added the equations for the evaluation matrices.

Round 2
